# Phenotypically Discordant Anomalies in Conjoined Twins: Quirks of Nature Governed by Molecular Pathways?

**DOI:** 10.3390/diagnostics13223427

**Published:** 2023-11-10

**Authors:** Lucas L. Boer, Eduard Winter, Ben Gorissen, Roelof-Jan Oostra

**Affiliations:** 1Department of Medical Imaging, Section Anatomy and Museum for Anatomy and Pathology, Radboud University Medical Center, 6500 HB Nijmegen, The Netherlands; 2Pathologisch-Anatomische Sammlung im Narrenturm-NHM, 1090 Vienna, Austria; 3Department of Medical Biology, Sections Clinical Anatomy & Embryology, Academic Medical Center, University of Amsterdam, 1105 AZ Amsterdam, The Netherlands

**Keywords:** conjoined twins, discordances, birth defects, anomalies, embryology

## Abstract

A multitude of additional anomalies can be observed in virtually all types of symmetrical conjoined twins. These concomitant defects can be divided into different dysmorphological patterns. Some of these patterns reveal their etiological origin through their topographical location. The so-called shared anomalies are traceable to embryological adjustments and directly linked to the conjoined-twinning mechanism itself, inherently located within the boundaries of the coalescence area. In contrast, discordant patterns are anomalies present in only one of the twin members, intrinsically distant from the area of union. These dysmorphological entities are much more difficult to place in a developmental perspective, as it is presumed that conjoined twins share identical intra-uterine environments and intra-embryonic molecular and genetic footprints. However, their existence testifies that certain developmental fields and their respective developmental pathways take different routes in members of conjoined twins. This observation remains a poorly understood phenomenon. This article describes 69 cases of external discordant patterns within different types of otherwise symmetrical mono-umbilical conjoined twins and places them in a developmental perspective and a molecular framework. Gaining insights into the phenotypes and underlying (biochemical) mechanisms could potentially pave the way and generate novel etiological visions in the formation of conjoined twins itself.

## 1. Introduction

Conjoined twins are a rare phenomenon with largely unexplained etiology and prevalence of around 1.5 per 100,000 births [1]. However, prevalence numbers can vary profoundly among different types of conjoined twins and changing socio-economic and/or geographical stratification [2]. Prevalence data specific for different types of conjoined twins remain largely absent and make many conjoined twins that do not fit within commonly accepted morphological phenotypes unique but puzzling entities [3,4]. It is commonly accepted that conjoined twins are divided into asymmetrical and symmetrical phenotypes [5]. Asymmetric conjoined twinning is an extremely rare event, occurring approximately in 1 in 1,000,000 live births. These forms of atypical twinning are oftentimes interpreted and described as heteropagi or parasitic twins, indicating that the smaller and vegetative member parasitizes on the larger (often viable) autosite [6]. The topography of these parasites is in concordance with known forms of symmetrical conjunctions, creating the general assumption that asymmetrical twins originate from initially symmetrical twins [4]. However, in respect to parasites, development is presumably complicated by an embryonic disk size imbalance or disruptive incident causing developmental delay in one twin member [7]. Symmetrical conjoined twins can be divided into dorsal (bi-umbilic) and non-dorsal (mono-umbilic) twins, of which the latter can be subdivided into ventral, lateral and caudal conjunctions. However, these mono-umbilical conjoined twins often show overlapping ventro-lateral or caudo-lateral phenotypes, creating divergent variability and a heterogeneous spectrum of conjunctions [5]. The pathogenesis of symmetrical conjoined twins is still disputed but can potentially be etiologized by the formation of two—instead of one—embryonic primordia in a single embryoblast. These initial duplications ultimately initiate and form two embryonic disks in close proximity that subsequently interfere with different embryologic developmental fields. The type, severity and overall morphology of conjoined twins are determined by the original and mutual localization, and potential mutual interactions of the duplicated primordia [5,8].

The term ‘asymmetry’ requires some additional digression, as this term is multi-interpretable within the context of (phenotypical) discordances in (parasitic) conjoined twins [9]. First, asymmetry refers to a phenotypical discordance in autosite–parasite configurations. The degree of development and its parasitic organization is quite variable and ranges from unrecognizable tissue lumps to well-formed body parts up to more or less complete but malformed or underdeveloped bodies, creating the difficulty of distinguishing highly developed parasites from discordancy in malformations in symmetrical conjoined twins [10,11,12] (Figure 1). Secondly, within symmetrical conjoined twins, asymmetry also refers to discordances at the organ or more structural level. Finally, asymmetry is commonly used in the context of, e.g., ‘anterior’ and ‘posterior’ facial defects in cephalothoracoileopagus conjoined twins with lateral deviations (Figure 2).

Before one can start describing any concomitant anomaly in conjoined twins, one should be acquainted with the semantics that should be used for these dysmorphological patterns and should know how they ought to be interpreted, as observations and descriptions of concomitant anomalies are oftentimes misinterpreted [13] or rather superficially etiologized [14,15]. The first dichotomy is to divide anomalies by their topographical location. Anomalies located at the conjunction site should be interpreted as ‘shared’ and should thus be inherently differentiated from those that are situated outside these borders. These shared anomalies can be divided into two distinct groups with their own etiology. The first group consists of morphological adjustments in the conjunction area as a consequence of interaction aplasia and/or neo-axial orientation and thus only present in conjoined twins of particular sort and degree of conjunction. A well-known example is the holoprosencephaly-like morphology of the ‘posterior’ compound face in cephalothoracoileopagus with lateral deviations (Figure 2).

The second group does not always arise and only occurs within particular settings of primordial duplications. Examples are (cranial) neural tube defects in parapagus diprosopus and diaphragmatic hernia in ventrally conjoined twins. These two groups of anomalies are outside the scope of this paper and reported comprehensively elsewhere [3].

Anomalies located beyond the borders of conjunction—and thus intrinsically non-shared—are to be described and interpreted as concordant and/or discordant with the anomaly itself, the severity of the anomaly and its location. These three patterns crystalize into three phenotypically distinct groups: (1) Concordance in the anomaly and its respective severity: both twins show the same anomaly with the same degree of severity. (2) Concordance in the anomaly but discordance in its severity and/or location: both twins show the same anomaly, but its severity and/or location differs (Figure 3A). This group also includes the so-called mirror-image anomalies (Figure 3B). (3) Within the last group, discordant patterns that are only observed in one twin member are seen.

Although the etiology of concordant concomitant and mirror-image anomalies is still disputed, the focus in this paper lies in discordant anomalies, as they are among the rarest and most elusive phenotypes that can be found in conjoined twins. Conversely, one of the biggest drawbacks in describing true discordances is the fact that actual discordant anomalies are present in only a small percentage of conjoined twins, inherently making these phenotypes extremely rare [13]. On the other hand, it is described that associated major malformations occur in 80% of conjoined twins and that more than one out of five of these anomalies are discordant [16,17]. However, there is oftentimes no discrimination from anomalies likely acquired due to the twinning mechanism itself, indicating a possible over-reporting of apparent discordances. This premise also creates scientific unfamiliarity towards true discordant patterns, often leading to misinterpretation of novel cases. However, focusing on discordant anomalies could potentially give insights into different developmental pathways, as it is a fact that conjoined twins share identical intra-uterine and presumed genetic footprints and thus parallel developmental pathways. The cause of the unilateral appearance of anomalies in conjoined twins is, therefore, one of the biggest questions as to why and how they arose. This paper tries to place discordant phenotypes into a broader molecular and developmental perspective in order to shed some hypothetical and philosophical thoughts on the etiology of the anomalies itself but also potentially giving insight into the genesis of conjoined twins itself. As the ultimate phenotypical observations are laid down during early development, it is essential to philosophize about embryological processes. The main questions to answer are (1) which discordant patterns are known to occur in mono-umbilical twins and (2) which developmental processes could potentially gear, initiate or influence these particular phenotypes in a broader sense.

## 2. Materials and Methods

What follows is an overview of existing cases of discordances within different groups of mono-umbilical conjoined twins that could be retrieved from the current literature. Cases were harvested according to semi-structured search criteria (mesh terms/title searches such as conjoined twins, conjoined twin, additional anomalies, concordance, discordancy, concomitant, anomaly, anomalies including specific defects that are known to occur as a discordant phenotype in conjoined twins), using PubMed and Google (scholar) to search for cases of conjoined twins with genuine discordant patterns. As earlier attempts to systemically search for particular phenotypes were rather disappointing and indicated that not all reports could be harvested according to certain search terms, we applied the ‘snowballing technique’ to include novel cases. The older literature that was not readily available was excluded. Reports were re-evaluated and phenotypically re-diagnosed according to our own observations. Cases of asymmetry within severe parasitic regression and anomalies due to the twinning mechanism itself, such as holoprosencephalic-like anomalies in cephalothoracoileopagus and neural tube defects in parapagus diprosopus [3], were excluded, leaving true discordances that are not immediately traceable to twinning mechanisms the only cases that were included in this review. These results are supplemented with novel cases that were found during surveys and after visiting a number of medical collections throughout Europe, including University of Brussels (Belgium), *Narrenturm* in Vienna (Austria), *Collections d’anatomie pathologique Dupuytren* at Sorbonne Université in Paris (France), *Berliner Medizinhistorisches Museum der Charité* (Germany) and the anatomical collections of university medical centers in Nijmegen, Utrecht, Amsterdam and Leiden (The Netherlands), a source which is rather unfamiliar in terms of harvesting previously unattested cases of malformations. After inventorying the different discordances per union group, we correlate these within a molecular and developmental framework to shed some light on the etiopathogenesis of one-sided birth defects in conjoined twins. Furthermore, we briefly digress on the spectrum of mono- and dizygotic twinning mechanisms and their discordances, as it is assumed that conjoined twinning is a sequential entity within the spectrum of monozygotic twinning. Finally, we discuss axis formation, asymmetry during embryogenesis and unilateral birth defects to widen the etiological horizon of discordant anomalies in conjoined twins. This approach is, in our opinion, an important step to take, as developmental perspectives remain virtually absent when theorizing about discordant phenotypes in conjoined twins. However, they are essential to take into account, as it is a fact that many developmental pathways take the wrong track during embryomorphic formation. The questions of why, how and when these complex phenotypes might arise and occur can only be answered when a broader developmental perspective is taken into account.

## 3. Results

Within our semi-structured survey, we found 58 cases of discordances in conjoined twins that could be retrieved from the literature. An additional 11 cases were found after inventorying and visiting different medical collections throughout Europe. All cases are shown in Table 1.

### 3.1. Discordance in Laterally United Twins

Laterally united conjoined twins share vast parts of their body (abdomen, thorax, neck, face and/or head) and classically consists of two phenotypes: parapagus dicephalus (two heads) and parapagus diprosopus (two laterally oriented faces in a single compound head). One of the more striking and relatively often occurring discordances found in dicephalic twins is anencephaly, with at least seven described cases appearing in the recent literature [9,14,15,25,26,27,28,56]. These unilaterally occurring neural tube defects should be distinguished from concordant neural tube defects in parapagus diprosopus, which are the consequence of interaction aplasia, a phenomenon comprehensively described elsewhere [3]. In addition, a skeletonized parapagus dicephalus discordant for anencephaly was found at *Berliner Medizinhistorisches Museum der Charité* (Figure 4A), and a formalin-fixated specimen, in the *Narrenturm* collections in Vienna (not shown due to its severely macerated state). *Narrenturm* additionally houses a dicephalic discordant for cleft lip/palate (Figure 4B)—a peculiar phenotype that has been previously described [29]—and a dicephalic discordant for transverse limb deficiencies and club feet (Figure 4C). The latter seems to be the first one of this kind being known so far. Moreover, two previously described cases are dicephalic twins discordant for holoprosencephaly [30,57] (Figure 4D), a phenotype that is described to occur [31]. Finally, the *Vrolik* collection in Amsterdam houses a dicephalic fetal skeleton discordant for pre-axial polydactyly, a phenotype that appears to have been described only once [19].

In parapagus diprosopus, the more approximated axial structures create a spectrum in which two faces are present in one compound head. Rating (1933) reported a case of diprosopus that showed two separate eyes and a normal nose in the right face, and synophthalmia and a proboscis in the left face [30]. Diprosopus has been described with discordant cebocephaly [19], and a female diprosopus with right-sided cebocephaly and a left-sided face with unilateral anophthalmia and cleft lip palate [20], and a case with discordancy in cyclopia have also been described [21]. Furthermore, diprosopuses discordant for cleft lip [22] and discordant polydactyly [23] were retrieved from the literature. No additional cases were retrieved from our museal inventories.

### 3.2. Discordance in Ventrally United Twins

Ventrally conjoined twins consist of a gradual spectrum from superficially united twins up to twins deeply entranced throughout the entire body. The group of superficial ventrally united conjoined twins consists of three overlapping entities with increasing intimacy of their union: xiphopagus, omphalopagus and thoracoileopagus. All three are joined on the ventral aspects at the mid-ventral portion of the trunk, extending from the sternum to the umbilicus, and present two separate heads and seven or eight extremities depending on deviating axial angulations [8,33]. The literature revealed quite a few superficially united conjoined twins discordant for cleft lip and/or palate [8,13,24,34,35,36,37,58,59], cloacal anomalies [60], myelomeningocele [13,37,61], spina bifida [62], polydactyly [63], clubfoot [38], scoliosis with multiple vertebral anomalies [39] and anencephaly [42]. Moreover, unilateral imperforated anus and absent or rudimentary genitalia are known to have occurred in at least three cases [37,40,43]. Oostra et al. (1998) described cloacal exstrophy in one twin [19]. Chaurasia described a thoracoileopagus in which one twin showed scoliosis, an abnormal left auricle, short neck, varus deformity of both hands with a pedunculated thumb on the right side and a missing thumb on the left, together with phocomelia in the right upper limb. The other, somewhat bigger member showed bilateral club foot [10]. This case is again somewhat difficult to distinguish from a highly developed parasite. The same holds for a thoracoileopagus in which the smaller twin showed mild micrognathia and the larger twin showed right-sided hemi-facial bony hypoplasia [41]. Spencer described a thoracoileopagus that showed a discordant Treacher Collins-like phenotype [64]. In addition, Oostra et al. described a thoracoileopagus twin from the *Vrolik* collection in Amsterdam discordant for multiple congenital anomalies but concluded that maceration hampered the examination [19]. Within the *Narrenturm* collection, we found two thoracoileopagi discordant for cleft lip/palate (Figure 5A,B) and one that was discordant for sirenomelia (Figure 5D) [44]. This phenotype seems to have occurred more than once, as the anatomical collections at Leiden University (The Netherlands) house a very similar specimen [30]. A specimen from *Collections d’anatomie pathologique Dupuytren*—Sorbonne Université, Paris (France)—also revealed a thoracoileopagus discordant for cleft lip/palate (Figure 5C). Finally, within *Berliner Medizinhistorisches Museum der Charité* (Germany), a thoracoileopagus with discordant anal atresia, micromelia and polydactyly was found (Figure 5E), a phenotype that appears not to have been described before.

Besides more superficially ventrally united twins, a more pronounced union creates a gradual spectrum towards prosopothoracoileopagus—in which the neck and facial structures are more deeply entrenched—and cephalothoracoileopagus conjoined twins, which are united entirely throughout the facial area. These deeply entranced twins then show two (in)complete faces on opposite sides of a compound and a single head (Figure 6). Cephalothoracoileopagi are oftentimes affected by discordant cloacal anomalies accompanied by meningo(myelo)celes [3,8,19,45,46,47,48], a phenotype that is highly similar to the so-called OIES complex (Figure 7). Kim et al. described a cephalothoracoileopagus that phenotypically showed discordant sex, but chromosomal analysis revealed female (XX) chromosomes. Moreover, after histological analysis, the apparent testes were actually ovaries with oocytes and without tubules [49]. The *Vrolik* collection in Amsterdam houses a cephalothoracoileopagus with discordant postaxial polydactyly [19]. We additionally found a prosopothoracoileopagus discordant for cleft lip/palate in the *Narrenturm* collections (Austria) (Figure 8A) and a cephalothoracoileopagus discordant for a longitudinal limb deficiency (Figure 8B), a phenotype that—to the best of our knowledge—again, has not been described before. The literature revealed discordant pre-axial polydactyly [46] and discordant syndactyly in cephalothoracoileopagi [47].

### 3.3. Discordance in Caudally United Twins

Classically, ileoischiopagi are caudally united twins with two vertebral columns located in a 180-degree opposite position, sharing lower abdomen, pelvis and perineum, with each twin having four normal limbs and a set of divided and diverted genitalia. Ileoischiopagi always have four upper limbs and two separate hearts [8]. However, ileoischiopagi often show considerable latero-caudal variations, resulting in a composite lower-limb and peno-scrotal aplasia on the dorsal aspect of the conjoined pelvis, hence the names ileoischiopagus tetrapus (four lower limbs) and ileoischiopagus tripus (three lower limbs) (Figure 9A,B). Ileoischiopagus conjoined twins are described with discordant hydrocephaly [49], microcephaly [51], anencephaly [52], meningocele [53], and cleft lip and palate [50,51]. Furthermore, female pseudohermaphroditism accompanied by discordant gastroschisis and anencephaly has been observed [54]. Khan described an ischiopagus tripus in which the healthier-looking active pink neonate showed a gradual infra-umbilical broadening of the torso towards a smaller neonate that was thin, emaciated, cyanotic, microcephalic, with a small torso and feeble reflexes. Again, this case could be interpreted as being a highly developed parasite [55]. An ileoischiopagus twin from the *Vrolik* collection in Amsterdam with facial dysmorphic features and hygroma colli indicating a possible association with Down syndrome was described by Oostra et al. (1998). Interestingly, another case of ischiopagus tetrapus conjoined twins in which the bigger twin was pinkish with good activity and the second twin was smaller, cyanotic and tachypnoeic, showing micrognathia, low-set ears, webbed neck and a cleft palate, was described, and the authors concluded that it resembled Pierre Robin syndrome [65]. Both syndromic findings were, however, based on external dysmorphological characteristics only and could very well have been highly developed parasites. Structural chromosomal variants have not been described to occur in conjoined twins so far.

## 4. Discussion

The presence of discordant anomalies unrelated to the site of coalescence may intuitively be interpreted as coincidental without any etiological connection to the twinning mechanism itself. However, their existence testifies that certain developmental pathways are altered and that each follows its own route and fate. This paper gives the most complete overview of discordant anomalies present in symmetrical conjoined twins up to date. However, we only describe 69 specimens with some sort of phenotypical discordancy. On the other hand, this relatively small number of cases is not surprising, as phenotypical discordances remain extremely rare phenotypes. Despite the relatively small sample size of discordances per group of conjoined twins, some preliminary thoughts are interesting to mention. It appears that more superficially united twins show more discordant patterns in comparison to more deeply united twins. From a developmental perspective, this would make sense, as embryological duplications are potentially located too far from each other to influence other embryomorphic fields and are prone to follow their own development destiny, which potentially includes discordances as stochastic phenomena. Nevertheless, an important fact to mention is that thoracoileopagus and parapagus dicephalus, both relatively superficially united, are the two types of conjoined twins that are most common, with 42% of cases being thoracoileopagus, and 11.5% of the cases, parapagus dicephalus [1], possibly biasing these observations. No novel cases of discordances in caudally united twins were retrieved from the museal collections that were inventoried, which is not surprising, as ischiopagus conjoined twins are observed in less than 3% of cases [1]. It seems that parapagus dicephalus cases were affected by discordant anencephaly more often, with at least nine cases being described in this paper. This is in comparison to one case of, e.g., thoracoileopagus, to say the least intriguing. Conversely, thoracoileopagi were relatively often affected by cleft lip/palate, with 12 cases. In addition, each conjunction group shows discordant limb anomalies, which include polydactyly, syndactyly and peromelia. Intriguingly, sirenomelia occurred twice. Finally, cephalothoracoileopagus seemed to be relatively often affected by discordant OIES-like anomalies, which was not that clearly observed in the other types of conjunction.

Although we focused on gross external discordances, it is known that many conjoined twins show some sort of internal discordance at the organ level. Even though a number of these patterns could potentially be caused and influenced by the twinning mechanism, one aspect should be taken into account when thinking about discordant anomalies in conjoined twins: a single compound heart. Many—if not all—conjoined twins with compound hearts show a multitude of cardiac malformations. Although these defects are presumed to be caused by the twinning mechanism itself, they could potentially influence subsequent outgrowth or may even be a predisposition for discordances or initiate parasitic regression (see further down).

To widen our current knowledge and viewpoint in respect to (discordant) patterns in conjoined twins and correlate these with other developmental perspectives, multiple topics are interesting to recall here, including twinning and the occurrence of structural defects; unilateral birth defects in singletons and the molecular basis for embryological asymmetry, axis formation, morphogenetic fields, and hypoxia; and the influence of hemodynamics on embryological development. As we feel that these themes are quintessential to take into account when theorizing on the genesis of conjoined twins and to link different types of expertise, these different topics are described below separately.

### 4.1. The Twinning Dogma and the Occurrence of Structural (Discordant) Defects

As it is generally assumed that conjoined twinning lies in the same spectrum of monozygotic twinning, we briefly highlight the currently considered concepts of twinning. Nonetheless, it appears that the common knowledge of twinning mechanisms is at best inadequate [66,67]. A fundamental certainty in the literature on twin biology and generally accepted—virtually intangible—perception is the acquaintance that monozygotic (MZ) and dizygotic (DZ) twinning events occur due to unrelated mechanisms. MZ twinning occurs due to the ‘splitting’ of the zygote, and DZ twinning starts with separate ovulations that yield parallelly formed embryos with no expectation of departures from a singleton outcome [68]. However, surprisingly, there is no evidence of any sort that a spontaneous human twin pair originates from two oocytes [67]. Moreover, it is oftentimes brought up as a fact that DZ twins are dichorionic and that monochorionicity is exclusively seen within MZ twins. Nonetheless, observations of monochorionic DZ twins [69,70] with discordant sex [71,72,73] contradict the accepted principle that monochorionicity is proof of monozygosity and opposes the current model of MZ and DZ twinning. Another issue when thinking about zygosity and chorionicity rises from the often absent information on the determination of zygosity in same-sex twins, as some identical-sex MZ twins could actually be DZ twins, again challenging the current model of twinning [74]. Implantation of a blastocyst with two inner cell masses is suggested as an explanation for MZ dichorionic gestations [68].

Another commonly accepted phenomenon in twinning is the increased incidence of congenital anomalies in comparison to singletons [75,76,77,78,79]. A large study including more than 17 million singletons and more than 380,000 twin births demonstrated an overall odds ratio for congenital anomalies in twins of approximately 1.3. Data suggest that the congenital anomaly rate in DZ twins is not significantly greater than that in singletons, but the rate in MZ twins is around 3–5 times greater [80]. This observation suggests that the required processes to generate MZ twins form such a disruptive event that asymmetry and, subsequently, discordances preponderate. Indeed, the risk of congenital anomalies is two times higher in monochorionic than dichorionic MZ twins, implying that chorionicity is an important factor in the origin of congenital anomalies [75]. Furthermore, it is assumed that many malformations find their origin in (intra-placental) vascular disruptions and/or pathophysiological changes [81]. The general concept that an excess of structural defects occur in MZ twins compared with DZ twins or singletons was first reported by Schinzel et al. [78]. However, hardly any of the twin pairs they included were assessed for zygosity. Instead, they were sorted into same-sex versus opposite-sex twin pairs, assuming that boy–girl pairs axiomatically concern DZ twins [82]. Interestingly, most—if not all—articles concerning twins base the dichotomy between MZ and DZ twins on this postulate. It is put forward that there is no reason to imagine that the cellular processes of embryogenesis in DZ twins are any different from those in MZ twins [67]. It was postulated that the double-ovulation hypothesis of DZ twinning is untenable, as this is purely a hypothetical model, and one should seek its explanation in syngamy and zygosis assembled to ‘*two zygote nuclei instead of one within the confines of the single secondary oocyte and its zona pellucida*’. The cells may all contain copies of one zygote nucleus then forming MZ twins. When syngamy yields two genetically distinct zygote nuclei with two distinct sibling cell lines DZ twins occur. (DZ twins). It is hypothesized that MZ and DZ twinning processes have the same list of consequences of anomalous embryogenesis and that both arise and differentiate from a single continuous mass of cells [68,82]. It should be noted that twinning remains a complex phenomenon and that many elements of the underlying processes, including the cellular processes of conception, polar body twinning, embryogenesis in MZ and DZ twinning, chorionicity, and finally chimerism and mosaicism, are still not completely understood [66,67,83]. Conventionally, it is supposed that MZ twins are genetically identical, ascribing successive phenotypical discordances to environmental influences that might subsequently alter and modify the expression patterns of the otherwise identical genetic endowment (shared or non-shared) [84]. However, recent insights indicate that this explanation is far too simple [85] and that genetic divergences due to post-zygotically induced point mutations actually occur [86]. A review by Gringras and Chen (2001) described genetic alterations in MZ twins and found that epigenetic modifications such as methylation, non-random or skewed X-inactivation, heterokaryotypical divergence and chromosomal mosaicisms can induce discordances in MZ twins [84,87]. Additionally, post-fertilization events such as post-zygotic non-disjunction in one co-twin and imprinting mechanisms [83,88] can cause discordances in MZ twins. It is known that divergent epigenetic variations can lead to different expression profiles of hereditary-disease genes [77,89]. Phenotypic discordances in MZ twins may partially be caused by de novo variations in copy number variants and/or mosaicisms [90]. Copy number variants are a major portion of the entire genome and are relatively unstable and strongly polymorphic, with mutation rates 100 to 10,000 times higher than those for single-base substitutions. Additionally, uneven exchange of cells during conception potentially leads to discordant feto-maternal microchimerism, possibly causing discordances in MZ twins [84]. It is becoming evident that spontaneous chimerism occurs far more frequently than previously thought [66]. Studies report MZ twins that are discordant for gender, chromosomal anomalies, single-gene mutations (both of nuclear and mitochondrial DNA), Mendelian disorders and other more structural anomalies [80,83,91,92,93,94]. There is some evidence suggesting that twinning processes may be associated with higher risk of congenital anomalies due to developmental disruptions that cause susceptibility to environmental agents. Moreover, dynamic interactions between stochastic and epigenetic variables potentially gear discordances in MZ twins [95,96]. It can be stated that MZ twinning itself can be regarded as an abnormality of morphogenesis that could potentially cause sequential anomalies [97]. Early primary malformations potentially develop due to overlapping etiological mechanisms and often predilect midline structures (e.g., sirenomelia, cloacal anomalies, neural tube defects and holoprosencephaly) [98]. It becomes clear that MZ twins are far from identical and that differences between ‘identical’ twins do occur, be it in a phenotypical sense or a molecular sense. This proposition creates the contradiction that identical intra-uterine environments can produce discordances. The question remains if this is also the case in conjoined twins, where there is one duplicated entity within a single amniotic sac and single umbilical cord. From this perspective, forked umbilical cords in monoamniotic twins with growth discordances [99] and mirror-image anomalies [100] are—to say the least—quite intriguing. As it is assumed that conjoined twinning lies on the same etiological horizon as MZ twins, one could argue that MZ twinning should be interpreted as a congenital anomaly in itself in which concomitant (discordant) anomalies are potentially causally related to different phenotypical patterns in conjoined twins. Indeed, overlap exists in structural defects that are both present in or between twins and in conjoined twins.

### 4.2. Unilateral Birth Defects in Singletons: Discordances in Left–Right Patterning?

A study by Paulozzi and Lary showed the lateral allocation of 6390 unilateral defects. Of the 102 ‘defect categories’, 57 showed an excess of defects on the right side of the body. Moreover, 39 defects had an excess on the left side of the body. Finally, six defects were equally distributed [101]. Paulozzi and Lary concluded that changes in the lateral distribution of specific birth defects may be caused by subtle differences in morphogenesis on the left and right sides of the embryo after establishment of left–right asymmetry, events that occur prior to organogenesis. The finding that more defect categories were right-sided than left-sided could be supported by the observation that mitochondrial development in rat embryos is delayed on the right side. The right side, therefore, may be more vulnerable to defects caused by prenatal hypoxia than the left one. It has been shown that alterations in mitochondrial metabolism influence cell differentiation and growth [102]. Interestingly, patients affected by Holt–Oram syndrome—which includes upper-limb malformations—show more defects in the left arm than the right one [103], while, e.g., fibular a/hypoplasia is more common on the right side [104]. Moreover, polydactyly is often observed as a unilateral anomaly [105]. It has been shown that both thumb and hallux duplications are mostly unilateral and predominantly involve the right foot [106]. Numerous disorders only affect one side of the body, including, among many others, craniofacial clefts, hemihyper- and hemihypoplasias, hemifacial microsomia, Parry–Romberg syndrome, hemimaxillofacial dysplasia, Klippel–Trenaunay syndrome, Sturge–Weber syndrome and Proteus syndrome. Apparently, some malformations tend to have a significant bias for one side or are phenotypically expressed unilaterally. An excellent review covering this topic is given by Cohen [107]. The fact that many anomalies tend to affect one side of the body is intriguing and interesting to take into account when discordant patterns within mono-umbilical twins are observed.

### 4.3. The Molecular Basis for Asymmetry

The left–right asymmetry of the vertebrate body is derived by a controlled motile-cilia-driven leftward flow in the transient left–right organizer called the node. This chiral rotation produces a vectorial leftward flow of extracellular fluids, eventually resulting in left–right asymmetric gene expression and, with that, developmental pathway differences [108,109]. Strikingly, most of these nodal cilia rotate in a clockwise manner, inducing a leftward fluid flow. A cascade of asymmetric gene expression during diverse developmental stages results in differential cell proliferation, migration and adhesion [108,110]. In recent years, several dozens of genes that are differentially expressed on the two sides of vertebrates have been found and include the transcription factors, growth factors and TGF-β signaling molecules that are responsible for left–right asymmetry during embryonic development [111]. Asymmetry can be broken down in three stages: (1) initial determination of the polarity (left or right); (2) expression of signaling molecules that establish such polarity; and (3) asymmetric morphogenesis induced by these signaling molecules [112]. The above-mentioned stages create effects through which embryomorphic fields become self-reinforcing to create asymmetric patterns of gene expression over much larger areas [108]. Interestingly, mutations in these fields can obliterate asymmetry, then implying mirror-symmetrical anatomy. In respect to this paper, laterality defects and mirror-image anomalies in conjoined twins are intriguing [113,114]. In every stage during development, cells in an embryo are presented with a limited set of options according to the state attained. Cells that ‘travel’ along developmental pathways branch repeatedly. Each step is initiated by a choice, and its sequence of choices determines its ultimate destiny and its morphology [115]. To reduce this complex phenomenon, one can ask the following: how do two cells with identical genomes become two different cells? A cell does not have to receive an external cue to change. One phenomenon is asymmetric cell division. During this event, certain molecules are divided unequally between two daughter cells. These asymmetrically segregated molecules act as the determinant of one of the cell fates by directly or indirectly altering the pattern of gene expression within the daughter cell that receives it [116]. Another way to change cells is to expose them to different environments with, e.g., signaling molecules from neighboring cells, a phenomenon called lateral inhibition, again leading to cell diversification. Moreover, another strategy to change cells is the process of inductive interaction. In this process, a group of cells that are identical and own the same developmental potential become influenced by an external signal from a cell outside the group, which drives a number of cells from the group into a different pathway. Generally, the signal is limited in time and space, so that only a subset of the component cells, those closest to the source of the signal, take on the induced character. The concept of inductive interaction is mediated by molecules and diffusion and can eventually form morphogenetic fields that are driven by morphogens (see further down). However, as morphogens typically act over a distance of less than one millimeter and only govern relatively simple choices (e.g., initial specification), another—more complex—phenomenon during, e.g., embryological patterning and growth is needed: sequential induction. In this strategy, a series of local induction processes generate a multitude of cell types starting from only the few that were initiated with a morphogen. Sequential induction embroiders successive levels of detail and thus acts on a bigger scale, creating a progressively more complicated pattern [115]. In addition, cells can change over time in their responsiveness to developmental signals; a cell is programmed to respond differently according to its age and its past history. Besides external cues, a cell does not have to receive external inductors to change. Sets of regulatory molecules inside the cell provoke the production of another, so cells can autonomously go through a series of different states. Genes that control the timing of these cellular events can mutate, causing heterochronic phenotypes. When this occurs, cells behave as though they belong to another stage of development [116]. It becomes clear that the specific molecular details of the mechanism that governs the temporal program of development are quite mysterious and near incomprehensible to oversee. However, when relating embryological concepts to the early formation of conjoined twins, it becomes clear that many fields during the initiation or genesis of conjoined twins may influence each other and that many precarious events could be altered due to deviant molecular flows, gene expression, mislocalized cues or other mechanisms that are misplaced in both a spatial sense and a temporal sense. An important aspect in the formation of the ultimate phenotype of conjoined twins could be that the duplication of certain structures, say, the node with its motile cilia, is influenced by this duplication, which includes midline barrier deviations/overlap that might change left–right patterning and its ultimate outcome [113].

### 4.4. Early Patterning in Vertebrate Development: Axis Formation

Throughout the rapid diversification of lifeforms during the ‘Cambrian explosion’—about 540 million years ago—the earliest chordates appeared [117]. Remarkably, but not surprisingly, genes regulating early chordate development are conserved among all vertebrate species, and the expression patterns of these early genes are remarkably similar across the entire vertebrate subphylum [118]. Interestingly, only a small number of patterning mechanisms are repeated [115]. Within early embryonic development, the applicable spatiotemporal organization of signaling morphogens is a fundamental component to control axis formation and is of decisive importance in establishing organisms’ body plan. The embryonic body axes serve as positional structures for subsequent pattern embellishment. In spite of the high number of players shown to be compulsory in early embryonic development and to be involved in the numerous mechanisms regulating patterning and morphogenesis, it has been shown that only two initial signals—bone morphogenetic protein (BMP) and Nodal—act at the top of a cascade of regulatory and inductive signals and are sufficient to trigger all required downstream developmental pathways, ultimately leading to the arrangement of a fully differentiated embryo [119,120]. Both BMP and Nodal are multifunctional growth factors belonging to the TGF-β *superfamily*. BMP activity is required for the dorsoventral axes [121], whereas Nodal is related to antero-posterior (cranio-caudal) patterning [122]. Interestingly, Xu et al. induced a secondary axis by injecting Nodal and BMP mRNAs that were completely independent of the orientation of the primary axis in zebrafish. In most cases, axes fused in the cephalic region, while others were completely separate and were placed (anti)parallelly or perpendicularly to the antero-posterior axis of the primary embryo [119]. It becomes more and more clear that differential BMP signaling activates a diverse program of gene expression in a dose-dependent way, thus specifying distinct cell fates across larger fields [121].

Chen et al. applied multi-omics to reveal the molecular characteristics of thoracoileopagus conjoined twins; they found differences between the conjoined fetuses in respect to DNA methylome, transcriptome and exome and raised the possibility that the Wnt signaling pathway could be involved in the conjoined-twinning process [123]. Interestingly, injections of mRNA Wnt signaling pathway components such as *wnt8*, *Siamois* or *β-catenine* induced conjoined twins in *Xenopus* [124]. Besides BMP and Nodal, the Wnt family of secreted proteins plays crucial roles in embryonic axis formation [125], gastrulation, and neural tube closure and patterning [126,127]. When human stem cells treated with Wnt and Activin were grafted into chick embryos, it contributed to a secondary axis [128]. Finally, Wnt is closely related to the Shh pathways, which in turn play critical roles in primary cilia function and the induction of ventrally and dorsally identified genes and gradients, regulating embryonic morphogenesis [127]. A complete overview of molecularly induced duplications of organizing centers in mammals and non-mammals has been previously described [5]. It has to be noted that a complex array of signaling cascades play a role in coordinated axis formation. Although more and more genes, proteins and insights into molecular pathways become unraveled, the underlying mechanisms of axis formation are far from being elucidated completely. Nonetheless, the above-mentioned points create room for thought about the possible etiology of conjoined twins, as it is postulated that the duplication of early axial structures is responsible for the ultimate phenotype and spectrum of mono-umbilical twins [5]. The definitive phenotype of any conjoined twins can be considered the gateway to prospecting early patterns of embryogenesis. When one observes a multitude of axial duplications within full-term conjoined twins, it is a fact that two very early organizers and their inducers are present from the very beginning of development when axis formation is set and initiated. Two parallel pathways could both take their own path and induce discordances; how they (molecularly) influence each other in such a way that discordances can evolve remains, however, elusive.

### 4.5. Morphogenetic Fields and Morphogens

It is beyond the scope of this paper to go in depth in all principles and processes during embryogenesis, a field that is rapidly evolving with ever-growing complexity. However, when theorizing on discordant phenotypes in conjoined twins, one should take morphogenetic fields into consideration, as morphogenetic fields define developmental potency and are responsible for the ultimate morphological fate [129,130]. A morphogenetic field can be defined as an embryonic area in which cells can co-operatively form a separate structure induced by signals such as morphogens. Morphogens are molecules that act at relative long range and are secreted by sender cells. In turn, a concentration gradient is achieved and can be translated by cognate signaling pathways in receiver cells to create distinct cell fate domains activating transcriptional effectors [131]. These gradient molecules thus ultimately create the morphology, hence their name, morphogens. Besides inducers such as morphogens, extracellular inhibitors of signal molecules model the response from the inducer. It is becoming evident that a large number of developmental decisions are controlled by inhibitors rather than initial signal molecules [115]. The potency of morphogens is immense, as they organize the entire morphological pattern mechanism with astonishing precision in space and time [132]. Morphogenetic field theory implies that information is processed and communicated at substantial distances across the developing organism and that the property of a field is its non-locality. An important aspect of the morphogenetic field and what information it encodes is the degree of modularity, and this tremendous potential was seen in the earliest transplantation experiments [133]. Moreover, it has been shown that fields are bioelectrically influenceable [134]. When amphibian embryos were exposed to externally applied DC current, field gradients were altered, and developmental defects preponderated, giving the assumption that organized bioelectric current is essential during morphogenesis [135]. Indeed, within the ‘Ion-flux’ model, asymmetric mRNAs pre-pattern the cleavage stage of *Xenopus* embryos and set up an asymmetric voltage gradient that, via gap-junction communication, drives the asymmetric localization of serotonin, which finally sets up the Nodal cascade [136]. Apparently, already in the four-cell stage of frog embryos, asymmetry preponderates and is initiated by pre-nervous signaling through neurotransmitters such as serotonin [137,138].

Why and how duplicated fields arise are among the biggest questions in respect to the formation of conjoined twinning. However, the ultimate phenotype gives away the fact that a multitude of fields are duplicated from the very start of embryogenesis. Interestingly, when a two-cell amphibian embryo is almost split into two by a hair loop, conjoined twins appear. When the splitting of the early embryo is performed so that two completely separate halves arise, two entirely separated and well-formed tadpoles are produced. The same holds after grafting a small group of cells (e.g., the dorsal lip of the blastopore in *Xenopus*) into a host embryo: gastrulation is initiated in both the graft and its own organizer. Although the initial cause remains unknown, it is imaginable that the duplication of particular embryomorphic fields induces duplicated concentration gradients. They are localized in a certain (pre)destined pattern, and both follow their own fate while inducing their own signaling pathways. It is known that defective developmental genes, differences in epigenetic state and/or complex (chromosomal) mosaicisms can act differently in multiple locations [77]. Subsequently, these signaling pathways could potentially interfere with each other and create dysmorphological phenotypes [24]. If the potential morphogenetic duplicates of early primordial concentrations and/or developmental fields persist, it is imaginable that mirror-image conjoined twins—and mirror-image concordances and discordances—might arise [139].

### 4.6. Effects of Hypoxia and Hemodynamics on Embryological Development

Low levels of oxygen (hypoxia) occur naturally and transiently during vertebrate development, driving vasculogenesis, angiogenesis, hematopoiesis and chondrogenesis [140]. However, experimental studies with episodes of moderate hypoxia led to developmental abnormalities or embryonic death. Transverse reduction limb anomalies are repeatedly associated with hypoxic insults during pregnancy, proposing that these defects may be the most evident markers of such events [141]. Furthermore, hypoxia plays essential roles in the formation of the developing nervous system, and non-physiological hypoxia disrupts development [142,143]. Additionally, hypoxia promotes emigration of neural crest cells [144]. It is described that severe hypoxic states can cause cleft lip and palate [145]. It becomes clear that hemodynamics play a paramount role in embryological morphogenesis and development [146]. During the first week of embryonic development, the vascularized portion of the yolk sac is the foremost gas-exchange organ. Possible consequences of yolk sac injury due to, e.g., diabetic pregnancies, include impairment of nutrient transportation due to vasculopathy. Recent studies reveal that the quality of yolk sac vasculature is inversely related to embryonic malformation rates [147]. In respect to discordant patterns within conjoined twins, it could be hypothesized that the duplicated embryonic disk with its single compound yolk sac is just not sufficient enough to nourish the entire embryo. On the other hand, the amount of blood needed to grow a complex duplicated organism could be just too great to accomplish in a number of times. It is oftentimes stated that parasitic twins occur due to hemodynamic instabilities or disruptions. Could vascular anastomosis, both intra- and extra-embryonically, be the cause of severe malformations? Twin reversed arterial perfusion syndrome is interesting to notice in this respect [148]. Furthermore, it has to be noted that the single umbilical cord in conjoined twins could potentially be prone to cause hemodynamic instability in one of the twin members and induce parasitic regression or potentially discordant patterns. Moreover, many conjoined twins with overlapping embryonic heart fields—and thus ultimately presenting with a compound heart—do show a large number of additional cardiac anomalies. Twins with two separate hearts often show discordant heart defects [149]. As a matter of fact, most—if not all—known congenital heart defects that can occur in singletons can be encountered in conjoined twins [150]. Within conjoined twins, abnormal, duplicated and triplicated in- and outflow tracts are known to occur [4,151]. These structural heart defects could potentially influence development due to insufficient flow and thus an altered nutrient/oxygen state. This, in turn, could hamper growth and potentially influence subsequent developmental pathways and theoretically initiate discordances. The abundant presence of congenital heart defects in many malformation syndromes is, in this respect, quite intriguing [152].

## 5. Conclusions

It is quite challenging to retrieve cases from scientific databases, as they are oftentimes interpreted and/or described erroneously, making it near impossible to search them systemically. As a matter of fact, only a very small percentage of cases are described comprehensively and are correct. Although no additional statistics were performed and our data merely represent a number of observed cases, it is tempting to speculate on certain aspects of our 69 described specimens. As it appears, some anomalies occur more frequently within certain types of conjoined twins (e.g., cleft lips/palates in thoracoileopagi and anencephaly in parapagus dicephalus) than one might expect. However, numbers are too small to suggest specific associations and could merely represent spurious associations influenced by stochastic events. However, we do feel that both ‘associations’ could be important observations when philosophizing on the genesis of conjoined twins. It becomes clear that the etiology of discordant patterns remains incredibly complex. When philosophizing about discordant anomalies, it is tempting to speculate whether discordant anomalies should be interpreted as being highly developed parasites, in which certain anomalies are (ostensibly) recognized as known congenital anomalies. One could hypothesize if a ‘developmental horizon’ exists, ranging from uni- and/or bi-laterally confined defects in symmetrical conjoined twins to complex associations and complete parasitic regression, which are potentially regulated by overlapping embryomorphic mechanisms, such as (micro) vascular disruptions, hypoxia or altered molecular or genetic propositions. Moreover, it remains unknown how and if duplicated left and right organizers, with their cilia dynamics, are influenced. It becomes more and more clear that it is essential to realize that conjoined twinning should be interpreted as a congenital malformation in itself that is secondarily influenced by abnormally united organs and superimposed effects of anomalous hemodynamics and molecular aberrations due to embryological and/or mechanical adjustments after twin formation [79,152,153]. Nonetheless, significant gaps still exist in our understanding of the exact mechanisms that initiate twin formation with or without additional (discordant) anomalies (see graphical abstract). Future approaches should not only focus on morphological and embryological expertise but likely entail a mixture of (molecular) cell biology and genetics. For now, it remains a fact that discordant anomalies in conjoined twins are still perceived as quirks of nature.

## Figures and Tables

**Figure 1 diagnostics-13-03427-f001:**
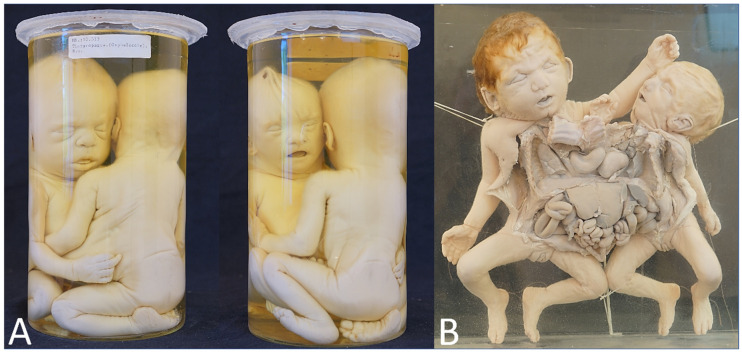
(**A**) Thoracoileopagus conjoined twins in which one of the heads is significantly smaller and clearly shows a different profile compared with the other. (**B**) Thoracoileopagus conjoined twins in which one member has a clearly smaller overall appearance than the other member (including different hair color). It can be questioned whether these phenotypes should be interpreted as discordant patterns or categorized as highly developed parasites. The latter—although debatable—seems more appropriate in these cases. Specimens from *Narrenturm* in Vienna (Austria).

**Figure 2 diagnostics-13-03427-f002:**
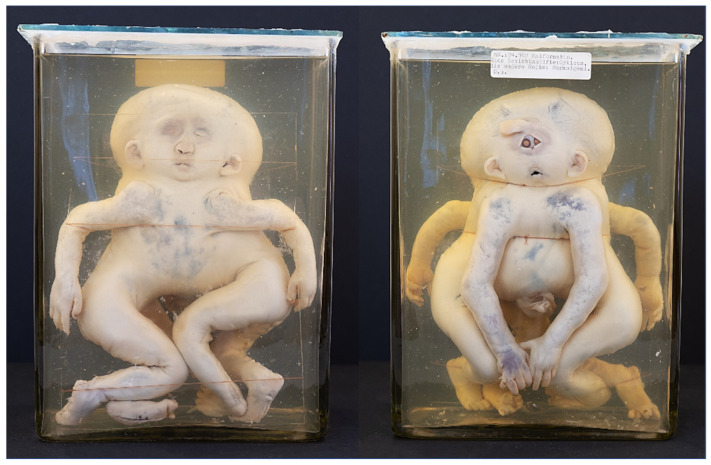
Cephalothoracoileopagus conjoined twins photographed from both sides with lateral—mostly cranially located—deviations resulting in a relatively complete compound face on the ‘anterior’ aspect (**left**) and diminished holoprosencephalic appearance of the ‘posterior’ face (**right**). Although this phenotype is clearly asymmetric and can be interpreted as being discordant, this particular phenotype is solely attributed to cranial interaction aplasia influencing the midline of compound structures. This shared twinning mechanism results in anomalies that mimic known patterns of malformations but should be interpreted as being part of the twinning malformation itself. This makes it sometimes difficult to distinguish true discordances from ostensible ones. Specimen from *Narrenturm* in Vienna (Austria).

**Figure 3 diagnostics-13-03427-f003:**
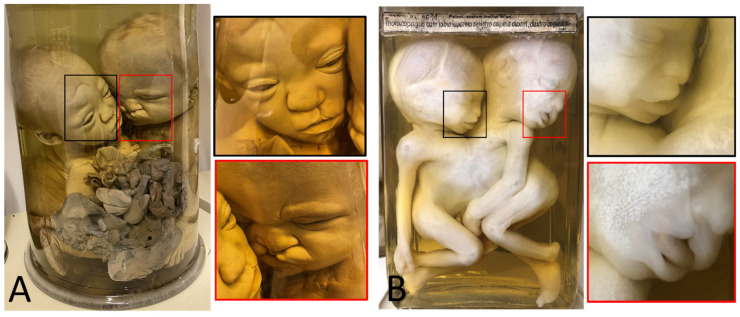
(**A**) Thoracoileopagus twins with concordance in facial clefts but discordancy in severity and location. One face shows a relative mild cleft lip (black), while the other face shows a more pronounced bilateral cleft lip/palate (red). Specimen from University of Brussels (Belgium). (**B**) Thoracoileopagus conjoined twins with mirror-image cleft lips. Specimen from *Narrenturm* in Vienna (Austria).

**Figure 4 diagnostics-13-03427-f004:**
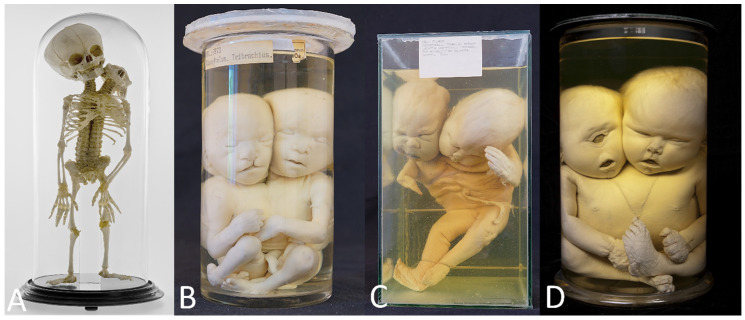
(**A**) Skeletonized dicephalic twins discordant for anencephaly. © Christoph Weber, *Berliner Medizinhistorisches Museum der Charité* (Germany). (**B**) Dicephalic twins with discordant cleft lip/palate. Specimen from *Narrenturm* in Vienna (Austria). (**C**) Dicephalic twins with discordant upper-extremity peromelia in the right twin and discordant club feet in the left twin. Specimen from *Narrenturm* in Vienna (Austria). (**D**) Dicephalic twins discordant for holoprosencephaly. Specimen from the anatomical collection at Leiden University (The Netherlands).

**Figure 5 diagnostics-13-03427-f005:**
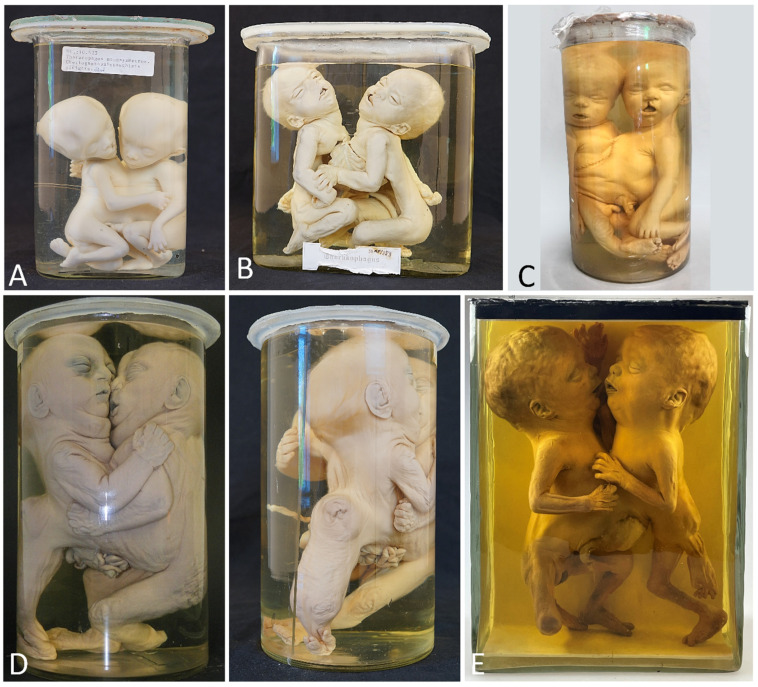
(**A**,**B**) Thoracoileopagus conjoined twins discordant for cleft lip/palate. Specimens from *Narrenturm* in Vienna (Austria). (**C**) Thoracoileopagus conjoined twin with discordant cleft lip/palate. Specimen from *Collections d’anatomie pathologique Dupuytren*—Sorbonne Université, Paris (France). (**D**) Thoracoileopagus discordant for sirenomelia. Specimen from *Narrenturm* in Vienna (Austria). (**E**) Thoracoileopagus with discordant anal atresia, micromelia and polydactyly from *Berliner Medizinhistorisches Museum der Charité* (Germany) © Navena Widulin.

**Figure 6 diagnostics-13-03427-f006:**
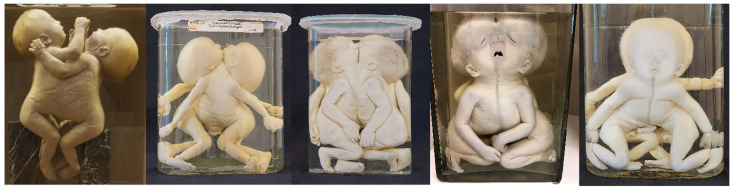
A gradual spectrum within ventral conjunction types. From thoracoileopagus (**left**) to prosopothoracoileopagus (**middle**) up to cephalothoracoileopagus (**right**) conjoined twins. Specimen on the left from Museum for Anatomy and Pathology at Radboud University in Nijmegen (The Netherlands), other shown specimens from *Narrenturm* in Vienna (Austria).

**Figure 7 diagnostics-13-03427-f007:**
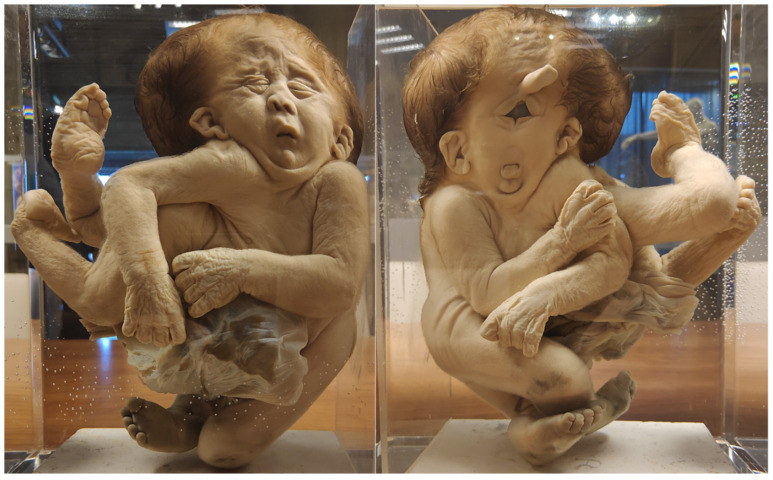
Cephalothoracoileopagus conjoined twins photographed from both sides in which—besides the holoprosencephaloid appearance due to lateral deviations in the cranial region—a discordant OEIS-like morphology is observed, with upward flexion of one set of lower extremities. Specimen from Museum for Anatomy and Pathology at Radboud University in Nijmegen (The Netherlands).

**Figure 8 diagnostics-13-03427-f008:**
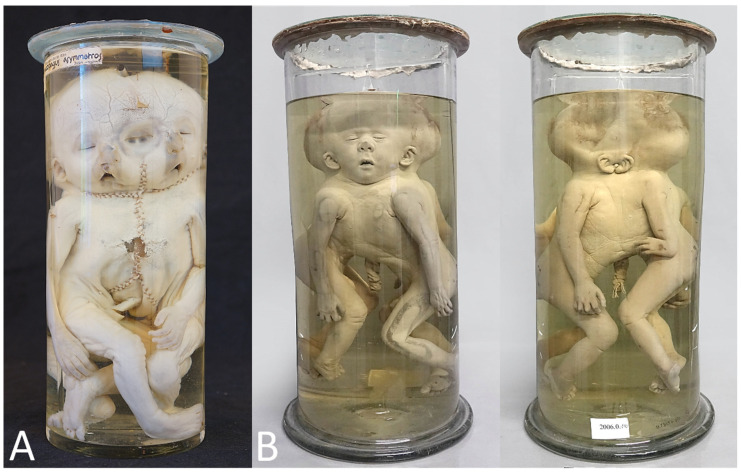
(**A**) Prosopothoracoileopagus conjoined twins with severe cranial deviations (creating two ventrally located faces as in parapagus diprosopus twins) discordant for cleft lip. Specimen from the *Narrenturm* collections in Vienna (Austria). (**B**) Both sides of a cephalothoracoileopagus conjoined twins with discordant longitudinal limb deficiency. Specimen from *Collections d’anatomie pathologique Dupuytren—Sorbonne Université*, Paris (France).

**Figure 9 diagnostics-13-03427-f009:**
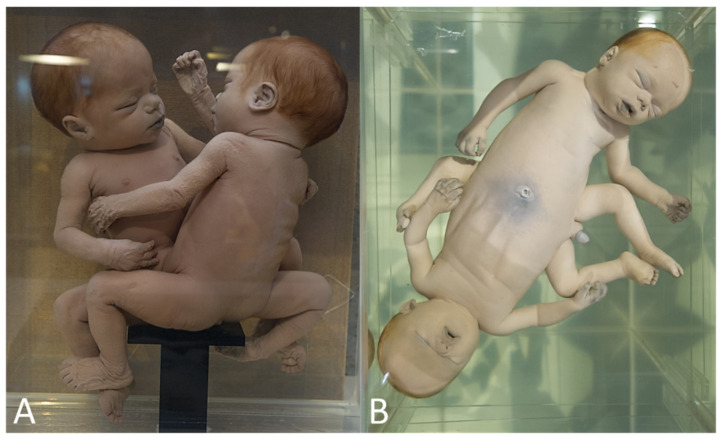
(**A**) Ileoischiopagus tetrapus. (**B**) Ileoischiopagus tripus. Specimens from Museum for Anatomy and Pathology at Radboud University in Nijmegen (The Netherlands).

**Table 1 diagnostics-13-03427-t001:** **Discordant anomalies in lateral, ventral and caudal conjunction patterns**.

Type of Twins with Discordances Retrieved from the Literature	Amount	Literature
Laterally united twins		
Parapagus dicephalus with discordant anencephaly	7	[9,14,15,18,19,20,21]
Parapagus dicephalus with discordant cleft lip/palate	1	[22]
Parapagus dicephalus with discordant holoprosencephaly	2	[23,24] Figure 4D
Parapagus dicephalus with discordant polydactyly	1	[25]
Parapagus diprosopus with discordant synophthalmia/proboscis	1	[26]
Parapagus diprosopus with discordant cebocephaly	1	[25]
Parapagus diprosopus with discordant anophthalmia and cleft lip palate	1	[27]
Parapagus diprosopus with discordant cyclopia	1	[28]
Parapagus diprosopus with discordant cleft lip	1	[29]
Parapagus diprosopus with discordant polydactyly	1	[30]
Ventrally united twins		
Thoracoileopagus with discordant cleft lip/palate	9	[8,13,31,32,33,34,35,36,37]
Thoracoileopagus with discordant scoliosis and vertebral anomalies	1	[38]
Thoracoileopagus with discordant anencephaly	1	[39]
Thoracoileopagus with multiple discordances in both twins *	1	[10]
Thoracoileopagus with multiple discordances in both twins *	1	[40]
Thoracoileopagus with discordant Treacher Collins-like features	1	[41]
Thoracoileopagus with multiple discordances *	1	[25]
Thoracoileopagus with discordant sirenomelia	1	[23]
Thoracoileopagus with discordant anal/cloacal malformations	3	[35,42,43]
Thoracoileopagus with discordant cloacal exstrophy	1	[25]
Cephalothoracoileopagus with discordant OEIS-like malformations	7	[3,8,25,44,45,46,47] (Figure 7)
Cephalothoracoileopagus with discordant polydactyly/syndactyly	3	[25,45,46]
Cephalothoracoileopagus with phenotypically discordant sex but genotypically female	1	[48]
Caudally united twins		
Ileoischiopagus with discordant hydrocephaly	1	[49]
Ileoischiopagus with discordant microcephaly	1	[50]
Ileoischiopagus with discordant anencephaly	1	[51]
Ileoischiopagus with discordant meningocele	1	[52]
Ileoischiopagus with discordant cleft lip and palate	2	[49,50]
Ileoischiopagus with discordant pseudohermaphroditism, gastroschisis and anencephaly	1	[53]
Ileoischiopagus with multiple discordances *	1	[54]
Ileoischiopagus with discordant hygroma colli and Down syndrome-like features	1	[25]
Ileoischiopagus with discordant Pierre Robin syndrome-like features	1	[55]
Total found twins with discordances from literature	**58**	
**Novel cases of Twins with Discordances Retrieved from Museal Surveys**	**Amount**	**Location**
Laterally united twins		
Parapagus dicephalus with discordant anencephaly	2	1 from Berliner *Medizinhistorisches Museum der Charité* (Figure 4A) and 1 from *Narrenturm*
Parapagus dicephalus with discordant cleft lip/palate	1	*Narrenturm* (Figure 4B)
Parapagus dicephalus with discordant transverse limb defects	1	*Narrenturm* (Figure 4C)
Ventrally united twins		
Thoracoileiopagus with discordant celft lip/palate	3	2 from *Narrenturm* (Figure 5A,B) and 1 from *Collections d’anatomie pathologique Dupuytren* (Figure 5C)
Thoracoileiopagus with discordant sirenomelia	1	*Narrenturm* (Figure 5D)
Thoracoileopagus with discordant anal atresia, micromelia and polydactyly	1	*Berliner Medizinhistorisches Museum der Charité* (Figure 5E)
Prosopothoracoileopagus with discordant cleft lip/palate	1	*Narrenturm* (Figure 8A)
Cephalothoracoileopagus discordant for longitudinal limb deficiency	1	*Collections d’anatomie pathologique Dupuytren* (Figure 8B)
Caudally united twins		
None		
Total found twins with discordances from museal collections	**11**	

* See text and original reference for all details.

## Data Availability

No new data was created.

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
