# Peer review of "Phenotypically Discordant Anomalies in Conjoined Twins: Quirks of Nature Governed by Molecular Pathways?"

_diagnostics, 2023, doi:10.3390/diagnostics13223427_

Round 1

Reviewer 1 Report

Comments and Suggestions for Authors

COMMENTS TO AUTHORS

Introductory paragraph: The authors present an interesting paper on 69 cases of a sub-group of conjoined twins. They carried out a literature search that yielded 58 cases and described 11 additional unpublished cases from various museums. The authors then present a lengthy discussion of the subject. While the intent of the authors is commendable, the “materials and methods” section of their manuscript is poorly described and lacks relevant citations. Thus, there are significant methodological challenges that must be addressed. Please, see below my major and minor comments, as well as specific comments.

General comments:

Major points:

·       In the methods section, the authors do not state the databases (e.g., PubMed?) from which the searches were done. Again, they are silent on the terms or phrases they used for the literature search. The paper will be improved if the authors address these concerns and probably present a flowchart depicting the databases used, search terms employed, the total number of manuscripts encountered, how many were excluded from further analysis based on exclusion criteria, etc.

·       The authors state in the methods section that “Reports were re-evaluated and phenotypically re-diagnosed by our own observations”. This approach introduces a level of subjectivity to the observations made! Are there no published guidelines for evaluating symmetrical conjoined twins that the authors are relying on “our own observations”? An alternate would be that about three different people could evaluate these reports independently before a phenotypic determination is made based on the three reports. Even with that approach, you must account for intra- and inter-assessor variabilities.

·       The authors state in the methods section that “These results are supplemented with novel cases that were found during surveys in a number of medical collections throughout Europe, a source which is rather unfamiliar to harvest previously unattested cases of malformations”. The authors may want to add a reference/citation for these “medical collections” so that the reader can trace the source of this information.

·       Page 4 lines 150-152 to Page 5 lines 153-157: Again, the authors may want to cite published guidelines that were followed in the analyses described.

·       The manuscript will improve if the authors could estimate the prevalence of the different symmetrical conjoined twin types (lateral, ventral and caudal), as well as concordant and discordant anomalies in the entire dataset (58 published + 11 unpublished cases). Though the authors give descriptions of the various cases they encountered, just describing the concordant and discordant phenotypes in each case makes the manuscript tortuous to the reader. The authors may want to create a table showing different twin types, as well as concordant and discordant phenotypes. This helps to estimate the prevalence of the different twin types and phenotypes in the dataset.

·       The authors make too many assumptions and postulates in the “Discussion” session that are difficult to substantiate with published scientific evidence.

Minor points:

·       Page 4 lines 141-142: What do the authors mean by “current literature” and “Older literature”? What constitutes “current literature” and “Older literature”?

·       Page 5 line 161: The authors state that “All cases are shown in Table 1”. However, the table was not available for my review.

·       There are minor grammatical errors, such as punctuation and word usage (examples given in my specific comments below) that must be corrected in the manuscript.

·       Page 18 line 700: The authors state that “numbers are too small to suggest specific associations”. However, the authors presented no association testing in their results. It is, therefore, out of the blue to see this concept showing up in their conclusion.

·       Page 18 lines 705-706: The authors state that “This paper tries to place these rare phenotypes into a molecular and developmental perspective which might be a small step in unraveling their enigmatic genesis”. While I appreciate the informativeness of the lengthy “discussion” section by the authors, the information presented is largely general and not specific to conjoined twins. Thus, “mechanical, molecular, and spatiotemporal misexpressions and/or aberrations” SPECIFIC for the conjoined twins of interest are largely lacking.

·       In general, the “conclusion” session is too lengthy and the authors may want to paraphrase and revise it to a maximum of 7 to 10 lines.

Specific comments:

Page 2 line 72: Replace “that” with “than”.

Page 3 line 90: delete “are”.

Page 4 line 127: replace “extreme” with “extremely”.

Page 4 line 147: Delete “included”.

Page 11 line 325: Replace “relative” with “relatively”.

Page 16 line 622: Replace “there” with “their”.

Page 17 line 668: Split “maybe” to “may be”.

Comments on the Quality of English Language

The English language requires minor edits, particularly punctuations.

Author Response

Reviewer 1:

COMMENTS TO AUTHORS

Introductory paragraph: The authors present an interesting paper on 69 cases of a sub-group of conjoined twins. They carried out a literature search that yielded 58 cases and described 11 additional unpublished cases from various museums. The authors then present a lengthy discussion of the subject. While the intent of the authors is commendable, the “materials and methods” section of their manuscript is poorly described and lacks relevant citations. Thus, there are significant methodological challenges that must be addressed. Please, see below my major and minor comments, as well as specific comments. Dear reviewer, thank you very much for this meticulous and appreciated feedback on our manuscript. I will go through your points separately below and highlight the changes in the main text with track changes.

General comments:

Major points:

  • In the methods section, the authors do not state the databases (e.g., PubMed?) from which the searches were done. Again, they are silent on the terms or phrases they used for the literature search. The paper will be improved if the authors address these concerns and probably present a flowchart depicting the databases used, search terms employed, the total number of manuscripts encountered, how many were excluded from further analysis based on exclusion criteria, etc. This is true. We did not perform a systematic review, and this was not our intention as we know from earlier attempts that this is virtually impossible to do when one is looking for certain rarely occuring phenotypes as many reports lack a proper description, diagnosis or even neglect certain (discordant) anomalies. Only a handful of papers can be retrieved trough a semi-structural search via e.g. PubMed. This does not mean that our search was completely random. Our main approach was to search for papers through snowballing that were initially found with Meshterms/title searches such as, conjoined twins, additional anomalies, discordancy’s, discordant anomaly, and by specific anomalies that we know can occur as a concordant/discordant pattern in conjoined twins such as, cleft lip, anencephaly, etc.etc. By doing so the few reports that describe a discordant anomaly in the appropriate manner appeared. By going through these papers other cases could be retrieved that did not appear from the initial search, indicating that their indexing is quit odd and hard to retrieve from databases. Going with the snowballing approach we identified 58 cases and we believe that this is a very complete overview of the existing cases as we found out that after a while no more novel reports appeared but the same articles that were already included appeared again. So Indeed it seems that our search is not very well structures but by spending lots of time searching in a semi-structural way a very complete overview of cases were found. We will highlight this in the methods section and we hope this approach can suit your expectations.

  • The authors state in the methods section that “Reports were re-evaluated and phenotypically re-diagnosed by our own observations”. This approach introduces a level of subjectivity to the observations made! Yes you could interpreted it like this but this is something which is the norm when describing particular phenotypes in a morphological manner. As three authors (LB, BG and RJO) are all three experienced anatomists with a focus on embryology we should morphologically interpret these cases. Please see our other cited papers (Boer et al., 2019/21/23) for our plea to describe conjoined twins in the correct manner with theories that are comprehensible with embryological development. Although I can imagine that this report is somewhat different than the norm, it places a number of phenotypes in a broader (maybe here and there philosophical) developmental perspective. However, we think this review is important to show to the medical community to become aware that many concordant discordant anomalies can be present and should be etiologically interpreted in a certain way and /or can be possibly initiated by a number of developmental concepts. We see this paper as a guidance to link certain expertise (anatomy, embryology, molecular development etc) together as for now papers on conjoined twins often remain very superficial. We feel that insights into early processes are necessary to eventually come further into their possible etiopathogenesis. Are there no published guidelines for evaluating symmetrical conjoined twins that the authors are relying on “our own observations”? Indeed, no evaluating protocols or guidelines are present. Dysmorphology is a field in which one observes something, thinks about, talks about it with peers and describe them in a certain way. The latter is also still an issue as no guidelines are present to describe anomalies in a certain way. An alternate would be that about three different people could evaluate these reports independently before a phenotypic determination is made based on the three reports. Even with that approach, you must account for intra- and inter-assessor variabilities. Please see our explanations above, we are behind our observations and qualified to say something about their morphology, no guidelines are present and I understand that what you describe would be the most ideal situation and scientifically the most correct one, however this approach is never done when describing certain phenotypes.
  • The authors state in the methods section that “These results are supplemented with novel cases that were found during surveys in a number of medical collections throughout Europe, a source which is rather unfamiliar to harvest previously unattested cases of malformations”. The authors may want to add a reference/citation for these “medical collections” so that the reader can trace the source of this information. We inclined the museums in which the surveys/visits were performed. Please see text accordingly.
  • Page 4 lines 150-152 to Page 5 lines 153-157: Again, the authors may want to cite published guidelines that were followed in the analyses described. As stated previously, no guidelines exist in describing morphological patterns. One of the arguments to write this paper was to give an overview of the encountered discordant phenotypes in mono-umbilical twins to show the scientific community their heterogeneous outcomes and its correct diagnoses and sketch their possible etiological background within a broader developmental perspective. Something which is not done before but concerning to us quintessential to better understand and etiologize these phenotypes.
  • The manuscript will improve if the authors could estimate the prevalence of the different symmetrical conjoined twin types (lateral, ventral and caudal), This paper describes discordant patterns in conjoined twins which are only very sporadically occurring and does not focuses on prevalence numbers of specific groups that are symmetric in most cases. as well as concordant we did not focus on concordant anomalies in this paper, please see Boer et al. 2021 (Characterizing the coalescence area of conjoined twins to elucidate congenital disorders in singletons) and discordant anomalies in the entire dataset (58 published + 11 unpublished cases). We do not feel inclined to talk about prevalence numbers in our findings/article as numbers are just too small. A number of discordant patterns we found are never described before, it would not make sense to even speculate about their incidence or prevalence numbers. All cases we describe should be interpreted as being ultra-rare with sometimes an incidence of N=1. We do however feel the right to associate certain findings such as anencephaly in parapagus dicephalus conjoined twins and cleft lip palate in thoracoileopagus twins as it looks that these additional anomalies are occurring more often than one might expect. Table 1 was not inserted but shows the amount of twins in each group with their respective anomaly. We think this gives a nice overview of our findings and is the most complete overview of discordant phenotypes up to date.

Though the authors give descriptions of the various cases they encountered, just describing the concordant and discordant phenotypes in each case makes the manuscript tortuous to the reader. The authors may want to create a table showing different twin types, as well as concordant (we did not included concordant phenotypes in this article but only discordant ones) and discordant phenotypes. This helps to estimate the prevalence of the different twin types and phenotypes in the dataset. Please see table 1 that shows an entire overview of our findings. This was unfortunately not included in the first draft. I have submitted this again within the MDPI system.

  • The authors make too many assumptions and postulates in the “Discussion” session that are difficult to substantiate with published scientific evidence. I can partially agree on this observation and I feel free to place this in perspective. We feel that the first step in understanding these rare discordant phenotypes is to give a complete overview of existing cases to see if there are certain associations within particular groups with particular discordant patterns. This is presented within the different groups and in table 1. As virtually no thoughts are currently postulated on the etiology of these defects we feel the need to make the reader/medical community (anatomists, embryologists, developmental biologists, geneticists, pediatricians etc) aware that discordant phenotypes in conjoined twins should be interpreted within a certain developmental framework and should be connected to early genetic, molecular and theories of early vertebrate development. This is actually the first time that one tries to etiologize and link these patterns within a developmental perspective.

We felt that only describe different  phenotypes is one of the major issues in other reports, no broadening of its hypothetical etiology are ever reported, making novel reports almost always case reports. As it is completely unknown how these discordant patterns arise, and we think that these patters come from very early defects during embryogenesis, we feel inclined that subjects such as axes formation (every conjoined twins does start with two axes), morphogenetic fields (it is a fact that conjoined twins show overlap and duplication of certain fields that could potentially interfere with each other), hemodynamics and asymmetrical gene expressions (to show the reader that asymmetry is a concept which can be used to etiologize unilateral birth defects and maybe also discordant patters), are important to notice. So indeed it can be interpreted that we make a number of associations/assumptions/postulates but we do feel that this paper is the start to become aware on deeper embryological concepts and- for now -the reader should be informed in a broader sense   

Minor points:

  • Page 4 lines 141-142: What do the authors mean by “current literature” and “Older literature”? What constitutes “current literature” and “Older literature”?
  • Page 5 line 161: The authors state that “All cases are shown in Table 1”. However, the table was not available for my review. I already noticed that MDPI did not enclose this in the draft paper. I have uploaded this together with the main draft of the paper during the reviewed submission and will highlight the importance of the table to be included as this will give the entire overview of found cases per conjunction group, the correct diagnoses and the PMID/museal reference.
  • There are minor grammatical errors, such as punctuation and word usage (examples given in my specific comments below) that must be corrected in the manuscript. We have adjusted this accordingly and also corrected some sentences that were retrieved to be changed from the editorial office. Please see track changes throughout the main text.
  • Page 18 line 700: The authors state that “numbers are too small to suggest specific associations”. However, the authors presented no association testing in their results. It is, therefore, out of the blue to see this concept showing up in their conclusion. Indeed we did find some things in our observations which are intriguing and worth mentioning. However, as too little numbers, with multiple variations are present for any statical analysis we did not perform any. However, we do feel to speculate about our findings. We have adjusted things in the conclusion however to makes this clear.
  • Page 18 lines 705-706: The authors state that “This paper tries to place these rare phenotypes into a molecular and developmental perspective which might be a small step in unraveling their enigmatic genesis”. While I appreciate the informativeness of the lengthy “discussion” section by the authors, the information presented is largely general and not specific to conjoined twins. Thus, “mechanical, molecular, and spatiotemporal misexpressions and/or aberrations” SPECIFIC for the conjoined twins of interest are largely lacking. This is correct and also a fact when it comes to developmental perspectives of conjoined twins, this is all unknown but we do feel that one should start somewhere and that this should be approached within a broader sense as development includes an array of different subjects that all could initiate something or sequentially within the formation of conjoined twins. Within our earlier published material we went from theorizing the gastrulation model of symmetrical conjoined twinning (Boer et al, 2019) to concordant anomalies that are caused by certain mechanical pathways (Boer et al, 2021) up to particular phenotypes that can be interpreted as parasitic regression within the gastrulation model (Boer et al, 2023). The current paper deals with phenotypes that are almost always interpreted as being unknown and very difficult to understand. However, the first step - concerning to us - is to create an entire overview of the appearing phenotypes and place them in a broader perspective that should include the subjects we describe. So you are right that it remains a hypothetical manner to link certain thoughts on the etiology of discordant phenotypes but we feel that this is the most logic first step to get deeper into these phenotypes. We sincerely hope that this can explanation meets with your approval as it is a fact that all the ins and outs of discordant phenotypes are still mere hypothetical assumptions. Therefore we cannot present any hard facts on this matter.

  • In general, the “conclusion” session is too lengthy and the authors may want to paraphrase and revise it to a maximum of 7 to 10 lines. We agree that the conclusion is somewhat long and we have shortened it dramatically. Please see main text accordingly.

Specific comments:

Page 2 line 72: Replace “that” with “than”. = adjusted accordingly

Page 3 line 90: delete “are”. = adjusted accordingly

Page 4 line 127: replace “extreme” with “extremely”. = adjusted accordingly

Page 4 line 147: Delete “included”. = we cannot delete this word, but we adjusted the sentence

Page 11 line 325: Replace “relative” with “relatively”. = adjusted accordingly

Page 16 line 622: Replace “there” with “their”. = adjusted accordingly

Page 17 line 668: Split “maybe” to “may be”. = adjusted accordingly

Reviewer 2 Report

Comments and Suggestions for Authors

The article is well structured and compiled in terms of narrative. It is appropriate to associate the article with literature.  However, as stated in the title, the connection part with molecular pathways is missing. It is appropriate to re-edit the article by making molecular association additions.

Author Response

Reviewer 2:

The article is well structured and compiled in terms of narrative. It is appropriate to associate the article with literature. However, as stated in the title, the connection part with molecular pathways is missing. It is appropriate to re-edit the article by making molecular association additions. Dear reviewer. Thank you very much for this appreciated feedback. This feedback was also given by reviewer number 1.

Although we could not agree more with this statement it is a fact that the molecular aspects of twinning are truly unknown and virtually non-existing within current scientific literature. This was actually one of the points to write this article to make the medical community more aware of early developmental processes that could possibly initiate or alter certain paths and create the dysmorphologies we encounter in a full term state. We consider that, after our earlier appeared works on conjoined twins, that we should sought its etiological explanation in a very early stage but need to start with a molecular and developmental perspective, and that discordant patterns could reveal hints about the more general groups in which no additional anomalies are present. In this article we try to place these morphological entities which are sometimes occurring only once within this broader perspective, be it in a rather superficial way. Unfortunately this is, concerning to us, the only way for now to look at this matter and we hope that you appreciate that the discussion points are there to link different fields together which is not done before in such a way and the first step in understanding these very rare phenotypes. I can say that our group is one of the very few that focus on the possible etiopathogeneses of conjoined twins in a embryomorphic sense. Please see our previously published material on this matter. I hope that with the changes that came from reviewing you can accept this approach.    

Reviewer 3 Report

Comments and Suggestions for Authors

Hello colleagues!

Excellent work, I have no comments on the content, but I do have comments on the clarity of the data presentation.

1. Why is the null hypothesis not described and a graphical abstract given?

2. It would be good to systematize the answer and conclusion section

Author Response

Hello colleagues!

Excellent work, I have no comments on the content, but I do have comments on the clarity of the data presentation. Dear reviewer, thank you very much for this comment. Please see your two points of concern discussed below.

  1. Why is the null hypothesis not described and a graphical abstract given? You are right that our hypothesis is not that clearly postulated. I think it can be describes as follows: It is a fact that discordant anomalies occur, we can observe this. We think that by looking at these discordant patterns we might find “the” or “a” key to better understand these specific patterns but maybe more important also get hints on the etiology of conjoined twins itself, both still completely unknown. We have inserted our hypothesis within the main text and discus our findings in the conclusion in a more appropriate and transparent manner. I hope this approach is okay for now.

In addition, I like the idea of a graphical abstract. We made a simple one in which the mechanisms we feel are important in twin formation are presented and all could influence each other and its ultimate phenotype. We inserted this in the main text after the discussion. Please see the image below. Perhaps this makes the reader aware that many developmental processes can hypothetically do something or being altered within the complex development of a conjoined twins.